# A data-driven approach for constructing mutation categories for mutational signature analysis

**Gal Gilad**[1], **Mark D. M. Leiserson**[2], **Roded Sharan**[1] *

**1** School of Computer Science, Tel Aviv University, Tel Aviv, Israel, **2** Department of Computer Science and Center for Bioinformatics and Computational Biology, University of Maryland, College Park, Maryland, United States of America

* roded@tauex.tau.ac.il

## Abstract

Mutational processes shape the genomes of cancer patients and their understanding has important applications in diagnosis and treatment. Current modeling of mutational processes by identifying their characteristic signatures views each base substitution in a limited context of a single flanking base on each side. This context definition gives rise to 96 categories of mutations that have become the standard in the field, even though wider contexts have been shown to be informative in specific cases. Here we propose a data-driven approach for constructing a mutation categorization for mutational signature analysis. Our approach is based on the assumption that tumor cells that are exposed to similar mutational processes, show similar expression levels of DNA damage repair genes that are involved in these processes. We attempt to find a categorization that maximizes the agreement between mutation and gene expression data, and show that it outperforms the standard categorization over multiple quality measures. Moreover, we show that the categorization we identify generalizes to unseen data from different cancer types, suggesting that mutation context patterns extend beyond the immediate flanking bases.

**Data Availability Statement:** Code and links to data can be found here: https://github.com/GalGilad/DDC.

**Funding:** This research was supported by a grant from the United States - Israel Binational Science

## Author summary

Cancer is a group of genetic diseases that occur as a result of an accumulation of somatic mutations in genes that regulate cellular growth and differentiation. These mutations arise from mutagenic processes such as exposure to environmental mutagens and defective DNA damage repair pathways. Each of these processes results in a characteristic pattern of mutations, referred to as a mutational signature. These signatures reveal the mutagenic mechanisms that have influenced the development of a specific tumor, and thus provide new insights into its causes and potential treatments. Originally, a mutational signature has been defined using 96 mutation categories that take into account solely the information from the mutated base and its flanking bases. Here, we aim to challenge this arbitrary categorization, which is widely used in mutational signature analysis. We have developed a novel framework for the construction of mutation categories that is based on

Foundation (BSF), Jerusalem, Israel (RS and ML). The funders had no role in study design, data collection and analysis, decision to publish, or preparation of the manuscript.

**Competing interests:** The authors have declared that no competing interests exist.

the assumption that the activities of DNA damage repair genes are correlated with the mutational processes that are active in a given tumor. We show that using this approach we are able to identify an alternative mutation categorization that outperforms the standard categorization with respect to multiple metrics. This categorization includes categories that account for bases that extend beyond the immediate flanking bases, suggesting that mutational signatures should be studied in broader sequence contexts.

This is a *PLOS Computational Biology* Methods paper.

## Introduction

Mutational signatures are characteristic combinations of mutation types (here referred to as categories) that arise from a specific mutational process. The analysis of the activity (here referred to as exposure) of these signatures in cancer provides insight to the active biological mechanisms that drive it. The inference of these mutational signatures is based on mutation category counts in a set of samples, acquired via whole-genome or whole-exome sequencing. A common way to do this inference is by employing non-negative matrix factorization (NMF) to express the mutation count matrix as the product of an exposure matrix (where each row represents signature exposures in the corresponding sample) and a signature matrix (where each row is a probability vector over mutation categories, representing a signature).

Current research in the field of mutational signatures typically categorizes mutations to 96 categories (referred to here as the *standard* categorization), taking into account the point mutation and a single flanking base on each side of it (6 base substitution classes and 4 possible flanking bases on each side) [1, 2]. This limited context ensures that mutation counts per category are not too sparse for downstream analysis, but may be too restrictive. Nevertheless, these categories served to discover state-of-the-art mutational signatures [3] as cataloged in the COSMIC database.

Several authors [3–5] considered also a wider context of two flanking bases on each side of the mutation, and analyzed mutational signatures using the resulting 1536 categories. Similarly, Shirashi et al. [6] modeled mutational signatures using independent "features" rather than categories, which allowed them to effectively consider two flanking base pairs on each side in their analysis. Nevertheless, these variants of the standard categorization were arbitrary (built on the mutation and 1–2 flanking based on each side) and were not derived systematically from data. Moreover, there is evidence in the literature for higher order mutation contexts [7].

Here we propose a data-driven approach for constructing a mutation categorization for mutational signature analysis. Our approach is based on the assumption that tumor cells that are exposed to similar mutational processes, show similar expression levels of genes that are involved in the underlying mutagenic processes. In particular, we focus on DNA damage repair (DDR) genes, as about one third of mutational processes with known etiology are associated with DNA repair deficiency [3]. The hypothesis that mutational signatures can be related to aberrant gene expression or alterations in DNA repair genes is well supported. For example, the deactivation of MUTYH gene in cancer patients is associated with a specific mutational signature [8–10]. In addition, Kim et al. [11] identify significant correlations

between DDR gene expression and the activity of multiple mutational signatures. We develop an algorithm to find a categorization that maximizes the agreement between mutation and DDR gene expression data, and show that it outperforms the standard categorization over multiple quality measures. We build a data-driven categorization (DDC) of mutations that considers up to three bases away from the mutated base. We show that our categorization leads to improved performance in signature discovery and expression correlation across multiple cancer types. We further compare the identified signatures to those cataloged in COSMIC and analyze their etiologies.

## Materials and methods

We consider a wide context for mutations, which consists of three flanking bases on each side of the mutated base. Representing a 7-mer mutation sequence by an 8-long character string (consisting of three 5' flanking bases, the mutated base, the new base, and the three 3' flanking bases) we aim to group mutations to 96 categories based on their sequence patterns. The identification and evaluation of mutation categorizations is based on the following measures of categorization quality (described in detail below): (1) agreement between mutation data and DNA damage repair (DDR) gene expression data, (2) reconstruction error of mutation count data, and (3) agreement between mutation data and gene expression information of cancer related genes. We attempt to find a categorization that optimizes (1), and then use measures (2)-(3) to evaluate the categorization obtained. Below we describe these measures in detail and how they are used for optimization and evaluation.

### Categorization quality measures

In all measures below we assume that we are given a mutation categorization into $M$ categories and observe a normalized count matrix $V$ of order $N \times M$ that reports the proportion of times a mutation category $i \in [1, M]$ was observed in sample $j \in [1, N]$ (Fig 1A shows an unnormalized count matrix $V$ of $N$ samples and $M$ arbitrary mutation categories). The (normalized) mutation count matrix is assumed to factorize $V \approx WH$, where $H$ is a matrix of mutational signatures (probability vectors over categories) and $W$ represents their sample-specific exposures.

To avoid overfitting, we perform *out-of-sample* evaluation that is applied both to the data sets that are used for learning a categorization and to the data sets that are used for the evaluation of the categorization. Each data set has two types of samples: (i) samples with whole-genome-sequencing (WGS) mutation information; (ii) samples with mutation count data from whole-exome-sequencing (WES) as well as gene expression data. First, the signature matrix is learned by factorizing the WGS mutation count matrix (using the Kullback–Leibler variant of NMF [12]; details on how the factorization order is determined are given below, see *Fitness and Selection*). We normalize the signature matrix to account for the difference between 7-mer distribution (representing mutation opportunities) of whole-genome and whole-exome sequences (see Mutation opportunity normalization section below). Then, exposures of the unseen WES samples are derived via a non-negative least squares (NNLS) computation [13]. In case the signatures are known in advance (taken from COSMIC [14]), only the final NNLS step is performed. To allow evaluation of these signatures for varying factorization orders, we compute the average prevalence (in terms of sample exposure) of each COSMIC signature in a data set and rank signatures from the most prevalent to the least.

**Agreement between mutation and gene expression data.**   Given an exposure matrix $W$ and an expression matrix $E$ over the same set of samples, $W$ and $E$ represent two views of the sample data and we can compare them using canonical correlation analysis (CCA) [15] (we use the Pyrcca [16] implementation of regularized CCA), which maximizes the correlation

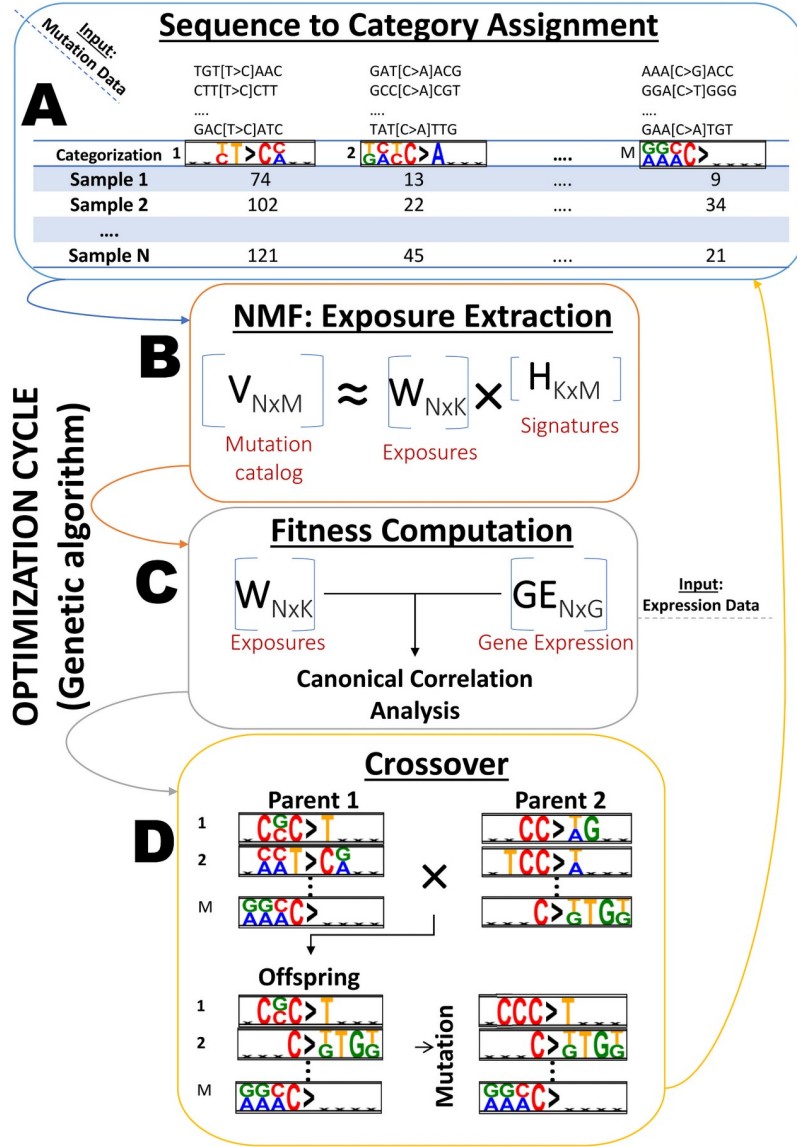

**Fig 1. Overview of the computational pipeline.** (A) For each mutation categorization (a set of *M* mutation categories), a mutation count matrix of *N* samples by *M* mutation categories is built through the assignment of 7-mer mutation sequences to their best matching category. (B) For each mutation categorization, its corresponding normalized mutation count matrix is factorized to produce a mutational signature matrix *H* and an exposure matrix *W* using NMF. (C) For each categorization, the correlation between the exposure matrix *W* and the corresponding gene expression data is computed using canonical correlation analysis to determine its fitness. (D) Categorizations are selected from the population for crossover and mutation to produce offspring for the next generation of the genetic algorithm.

between the two views. This correlation serves as a measure of the categorization quality. To avoid overfitting, we perform 10-fold cross validation, i.e., we learn the CCA coefficients using training samples and compute the resulting correlation on the test samples.

**Reconstruction error.** We compute the reconstruction error by applying NMF to the WGS samples to learn the signature matrix *H* and then derive the exposure matrix *W* from the other samples using NNLS. We then measure the approximation error (Kullback–Leibler divergence) of the factorization with respect to the test (normalized) count matrix *V*. To

minimize biases that arise from NMF's tendency to converge to local minima, we perform 10 runs with randomly initialized $W, H$ factorization matrices and report the smallest error obtained.

## Mutation opportunity normalization

Let $H$ be a matrix of mutational signatures inferred from WGS data. To account for the difference between mutation opportunities in whole-genome and whole-exome data, we first compute the counts of all possible 7-mer sequences in the GRCh37 genome and exome (genome and exon positions were downloaded from Ensembl: https://grch37.ensembl.org). Next, we construct two mutation opportunity 96-long count vectors, $U_{WGS}$ and $U_{WES}$, that correspond to the set of 96 mutation categories of $H$. Each position $i$ in those vectors is the number of 7-mer sequences in the genome/exome that map to the $i$th category. Finally, each row vector of $H$ is normalized by multiplying its $i$th element by $U_{WES,i}/U_{WGS,i}$ and normalizing the resulting vector so that it sums to one.

## Comparing our signatures to COSMIC signatures

Given a mutation count matrix for $N$ samples and all 7-mer mutation sequences, and a signature $S$ (probability vector over 96 mutation categories), we would like to compare its cosine similarity to a COSMIC signature. We construct a transformation matrix $T$ of size $96 \times 96$ from our mutation categories to the standard categories, where each $T_{i,j}$ is the fraction of 7-mer mutation sequences that belong to the standard category $j$ out of the total number of sequences that are mapped to our category $i$. Then, we compute the cosine similarity between $S' = ST$ and any COSMIC signature.

## Categorization optimization

We use a genetic algorithm [17] to optimize the mutation categorization. The algorithm maintains a population of possible solutions, where each individual in the population is a categorization. Each category within a categorization represents a pattern of mutations by a string of 8 characters that correspond to the three left flanking bases, the original base that mutated, the resulting base, and the three right flanking bases. Each symbol can be a single base or a combination of bases, except for the identity of the original base, which can be only C or T. Apart from this base, any other category symbol could represent any of the bases or any pair of bases (marked by a dinucleotide) or a wildcard base (marked by N), with the constraint that the mutated base is different from the original base.

Given a 7-mer mutation sequence of bases describing the mutation and its context, we assign this sequence to the category that best matches it based on three criteria (no mismatches are allowed): (i) the size of the category, i.e., the number of 7-mer mutation sequences that perfectly match it; (ii) the distance between the first and last non-N symbol of the category; and (iii) the distance of the farthest non-N symbol from the mutated site. These criteria are applied in order and used to resolve assignment conflicts.

For example, consider the 7-mer mutation sequence TGTCTAAC, which describes a C-to-T mutation and its context—TGT to the left and AAC to the right. Further consider the following mutation categories (using IUPAC code; e.g. S is G/C)): NSTCTANN and NNNCTARC. The 7-mer mutation sequence can potentially be assigned to both categories as there are no mismatches to both. We follow the three criteria to choose the best matching category: (i) both categories are of the same size—they cover 128 sequences; (ii) the distance between the first and last non-N symbol in each category is 5; (iii) the distance between the farthest non-N symbol and the mutated site is shorter in NSTCTANN, and so the sequence is assigned to it.

Categorizations are created and maintained in such a way that there are less than 1% sequence-to-category assignment conflicts (as described below). Each categorization includes a "joker" category (NNNNNNNN), which guarantees that all sequences are assigned to some category. For ease of comparison to the standard categorization, we focus on categorizations with $M$ = 96 categories.

**Ancestors.** Each category in a categorization is generated with a clearly defined (C or T) base in the fourth position, as part of an interval of $n\_symbols$ consecutive non-N symbols and whose starting position is chosen uniformly at random (see examples in Fig 2). Other non-interval symbols are assumed to be wildcards and the interval is constrained to contain the fourth position. We assume a range of 3 to 7 symbols for $n\_symbols$, with the highest concentration on $n\_symbols$ = 4—the number of bases in the standard categories, controlled by parameter $c$: $n\_symbols \sim Dir((1, c > 1, 1, 1, 1))$. For example, if $n\_symbols$ = 5 is generated, then an interval starting position is chosen randomly from the list [0, 1, 2, 3] of possible positions (guarantee a clearly defined base in fourth position). Then, if position 2 is generated, then a sequence of 5 symbols is generated randomly, with the constraints that the second symbol (corresponds with the fourth position—the original base) is C or T, and that the third symbol (corresponds with the mutated base) does not intersect with the original base. If TCRGA is generated, then this example results with the category NNTCRGAN.

**Fitness and selection.** A pre-step for fitness evaluation is the assignment of 7-mer mutation sequences to categories for each categorization, according to the criteria explained above, which results in a mutation catalog of size $N$ samples by $M$ categories (See Fig 1: Sequence to category assignment). In case of an ambiguous assignment, conflicting categories are replaced with new categories in order to resolve the conflict, until a rate of less than 1% ambiguous assignments of sequences to categories is achieved. The remaining conflicts are resolved by randomly assigning the sequence to one of its "best" matching categories. We now describe

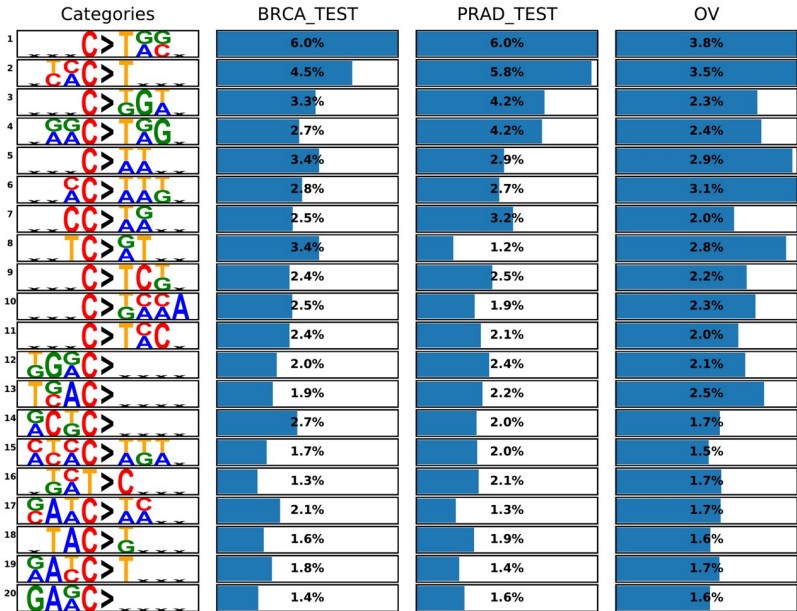

**Fig 2. Top categories and the percent of data set sequences that are mapped to each of them.** In each category, the bases at the fourth and fifth positions represent the mutation. Flat Xs represent wildcards. The top 20 categories with the highest average prevalence (i.e., the number of sequences that are mapped to a category divided by the number of data set sequences) over all 3 data sets are shown. Each bar is scaled from zero to the highest prevalence value in its column (data set).

the calculation of fitness scores for each categorization in the population. Let $K^*$ be the optimal number of NMF factorization components, determined by applying our CV2K method [18]. CV2K automatically chooses $K^*$ from data, based on a detailed comparison of the performance of different solutions (using different $K$ values) in reconstructing hidden values of the original mutation catalog in a cross validation setting, combined with a parsimony consideration. We focus on the range $K\_range = [K^* - 2, K^* + 2]$ of components. For each $K \in K\_range$ and each test fold, we compute the CCA correlation between DDR gene expression information and sample exposures as explained above. We take the mean correlation over all $K$ values and test folds to be the fitness score of a categorization.

Following fitness assessment, categorizations are ranked from 1 (worst) to the size of the population $S$ (best) and selected for crossover and mutation processes with probability $P_{selection} = \frac{rank_i^{power}}{\sum_{j=1}^{S} rank_j^{power}}$, without replacement. The *power* parameter enables us to adjust the strictness of the selection, i.e., the higher the value the more likely top ranked individuals are to be selected. The two categorizations with the highest fitness values proceed as-is to the next generation, to guarantee non-decreasing top fitness value. To perform the optimization on multiple data sets, we apply the above ranking system with respect to each data set, and construct an average ranking [19], i.e., an average over all data set specific rankings. We consider the individual and average rankings equally in the selection of categorizations for crossover. Finally, the best categorization in the population is defined as the top ranked categorization in the overall (average) ranking.

**Crossover and mutation.** We perform crossover and mutation in the population to produce the remaining offspring for the next generation. To produce a single offspring, we draw a categorization (parent) from the population with probability $P_{selection}$ (see above) and cross it with probability $q$ with another parent, or mutate it with probability $1 - q$ to produce an offspring. In case of a crossover, we randomly draw a different categorization (the other parent) from the population. A random percentage, up to 40%, of the categories are taken from the lower-fitness parent and the rest are taken from the higher-fitness parent, to produce the offspring. Note that if there are not enough unique categories in the two parents to create the offspring, we generate additional categories to fill up the categorization. Mutations occur after crossover is completed (or skipped), and are applied to each category with probability $m$. As category replacements naturally occur across the generations (either to resolve identified conflicts, or to fill up an offspring categorization), the mutation is responsible for more delicate changes—it replaces a symbol in a random category position with a different intersecting symbol, in order to expand/reduce the category with respect to this position (from nucleotide to dinucleotide symbol or the opposite transition, e.g. 'A' – > M ('A/C'), K ('G/T') – > 'G'). Consequently, a mutation from one symbol to a non-intersecting symbol can occur only indirectly, by a series of two or more mutations in a specific position. This approach was chosen to allow delicate changes in a categorization, while minimizing the introduction of new conflicts to it.

**Parameters.** We initialize the genetic algorithm with 100 categorizations (ancestors), 40 of which are selected according to their fitness, and a population of 40 individuals is maintained across the generations. We ran the algorithm for 500 generations (two days on 40 CPUs —Intel(R) Xeon(R) CPU E5–2699 v3 @ 2.30GHz). The genetic algorithm has four main parameters: crossover rate ($q$), mutation rate ($m$), selection strictness (*power*) and category concentration ($c$). To tune each of the four parameters, we set the values of the other three parameters and test the performance of the genetic algorithm (i.e., categorization fitness) on a range of values for the parameter in question, in order to find a value that avoids local minima and leads to greater fitness and faster convergence. For crossover rate, we examined the values 0, 0.4, 0.8, 1 and chose $q = 0.8$. For mutation rate, we tested rates 0.01, 0.05, 0.1, and set the

parameter to $m = 0.05$. For strictness of selection, we examined values 3, 5, 7 and chose *power* = 5. For category concentration, we examined values 1, 5, 10, 20 and set the parameter to $c = 10$.

## Data

We use the following TCGA [20] data sets that include gene expression and WES mutation data: (i) 976 breast BRCA samples that had been collected from BRCA-US project, available in ICGC [21]; (ii) 374 prostate PRAD-US samples; and (iii) 185 ovarian OV-US samples. In addition, we use WGS mutation data of: (i) 569 BRCA samples of the BRCA-EU project; (ii) 193 PRAD-UK samples; and (iii) 115 OV-AU samples.

For mutation data, we downloaded simple somatic data files from ICGC, extracted all the single base substitutions and used GRCh37 reference genome to collect 7-mer sequences centered around each of the mutations (3 flanking bases on each side and the substitution that occurred). These 7-mer mutation sequences served as initial categories, with 24,513 of these categories represented in the BRCA data, 23,310 in the PRAD data, and 24,150 in the OV data.

For gene expression data, we collected the corresponding data files from ICGC. For each sample, we extracted the value under 'normalized read count' for each gene, to create gene expression data sets of 20,502 genes, for each data set. We consider two types of gene subsets: (i) DDR genes, and (ii) Cancer Gene Census (CGC) genes. For (i), we use expression data for 190 genes from the updated version of [22] that can be found here: www.mdanderson.org/documents/Labs/Wood-Laboratory/human-dna-repair-genes.html. For (ii), we use expression data of 680 COSMIC Cancer Gene Census genes that do not appear in the DDR gene set.

We further split the largest BRCA-US data set to 726 training samples, that are used to learn our categorization with the genetic algorithm, and 250 test samples, that are used for evaluation. Similarly, we split the PRAD-US data set equally to 187 training samples and 187 test samples. We reserve the OV-US samples solely for testing.

As a reference set of signatures we used COSMIC signatures that were found to be active in the investigated cancer types: 1–3, 5, 6, 8, 13, 17, 18, 20, 26 and 30. These add up to a total of 12 signatures in the BRCA data set, 3 signatures in the PRAD data set, and 3 signatures in the OV data set.

## Results

Despite an almost decade-long research into mutational signatures, the categorization of mutations still revolves around 96 categories that specify a mutation and its two flanking bases (one on each side).

Here we aimed to take a data-driven approach toward mutation categorization with the goal of maximizing the correlation between the activities of the processes that create these mutations (a.k.a. mutation signature exposures) and the expression of the genes that are involved in the mutational process (DNA damage repair genes). An overview of the computational pipeline is shown in Fig 1.

We applied our pipeline to identify mutation categories over three data sets: breast cancer (BRCA) samples, prostate cancer (PRAD) samples and ovary cancer (OV) samples. To identify a categorization that generalizes across data sets, we performed multi-objective optimization using the two largest sample sets—BRCA and PRAD—for training. We benchmarked our categorization against the standard categorization, which is based on the mutation that occurred and a single flanking base on each side, referred to below as the *standard* categorization, and also against a random categorization which assigns 7-mer sequences of mutations (incl. the mutation that occurred and 3 flanking bases on each side) uniformly at random to 96

categories. To evaluate the categorizations, we tested their performance on a BRCA and PRAD test sets (250 and 187 left out samples, respectively) and an independent test set from different cancer type (OV), all evaluated in an out-of-sample fashion (see Materials and methods).

The categories that make the resulting categorization were variable in size (i.e. the number of sequences in the data set that are assigned to them) and greatly deviate from the standard categories (Fig 2). Of the 95 non-joker categories, 75 have multiple flanking bases on either side of the mutation, and 41 of them have high specificity up to 3 bases away from the mutation. 13 of these 41 categories have the same base appearing in all three flanking bases, either by itself, or as part of a dinucleotide symbol. There are 48 categories with flanking base specificity in only one side of the mutation. The 'A/G' dinucleotide symbol appears 51 times in our categories, while the other five dinucleotide symbols appear 30–37 times. There are 51 categories with C as the original mutated base and 44 with T, but of the most prevalent categories (Fig 2) only one category has T as the original base. This is expected since C mutations are 4–5 times more frequent than T mutations in the WES data sets.

We measured the quality of the resulting matrix factorizations and the agreement with the expression patterns of DDR genes and a set of cancer driver genes from Cancer Gene Census (CGC). Indeed, previous studies identified correlations between several mutational signatures and some cancer drivers and acknowledged that the cause-effect relation between signatures and cancer drivers can be in either direction [23]. We also used a control set that is not expected to be related to mutational signature activities—odorant binding genes (GO:0005549)—which does not share genes with the DDR and CGC sets.

In terms of factorization quality, our categorization performed best across a wide range of number of components ($K \in [2, \max(k\_cosmic, K^*) + 2]$, Fig 3). For agreement with expression data, we computed correlations between the inferred exposures of each categorization and expression levels of cancer-related gene sets: DDR genes and CGC genes. In this latter evaluation we also included 'standard cosmic' exposures that are inferred using the COSMIC signatures (corresponding to the standard categorization) in the given cancer type (see Materials and methods). Overall, our categorization yielded higher correlation values compared to the others. The results are summarized in Fig 4. Both the DDR gene set and the CGC gene set produced a similar range of correlation levels (0.3–0.5 for non-random categorizations). In contrast, the control set yielded correlation levels of up to 0.15, with no clear advantage for non-random categorizations over the random categorization (S2 Fig). For higher $K$ values, the improvement in performance (correlation to DDR and CGC genes) over the breast cancer data is marginal. This possibly stems from the difference in the signatures' DDR association between the cancer types. Specifically, BRCA has 12 known associated signatures, 6–8 of

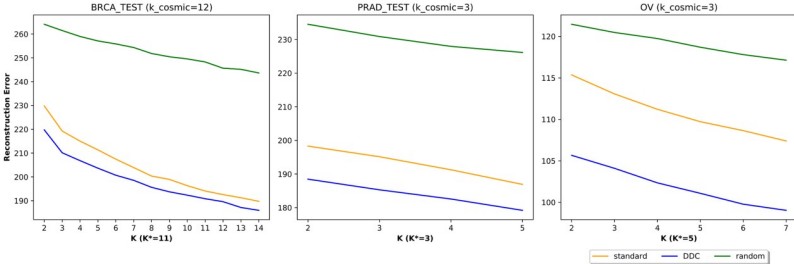

**Fig 3. Comparative performance evaluation by reconstruction error (log-transformed).** For each $K$ in the range of number of components $[2, \max(k\_cosmic, K^*) + 2]$, we apply NMF to the WGS samples to learn the signature matrix $H$ and then derive the exposure matrix $W$ from the test samples using NNLS. The reported reconstruction error (Kullback–Leibler divergence) is the approximation error of this factorization with respect to the test samples of the (normalized) count matrix $V$.

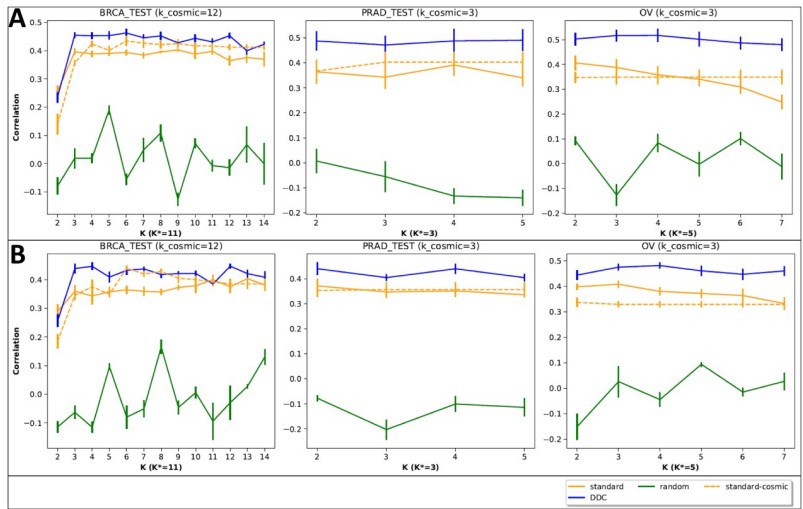

**Fig 4. Comparative performance evaluation: Correlation to expression of DDR genes (A) and CGC genes (B).** For each $K$ in the range of number of components [2, max($k\_cosmic$, $K^*$) + 2], we apply NMF to the WGS samples to learn the signature matrix $H$ and then derive the exposure matrix $W$ from the WES samples using NNLS. We learn the CCA coefficients using WES training samples and compute the resulting correlation on the test samples. The reported correlation is the average over 10-fold cross validation. Error bars represent the standard deviation of multiple evaluation runs.

which are believed to be associated with DDR, while both OV and PRAD have only 3 known signatures and 2–3 have some association with DDR [3].

To further examine our categorization and resulting signatures, we (i) computed the similarity between each of the signatures that were extracted using our categorization and all known COSMIC v2 signatures (including signatures believed to be associated with sequencing artifacts according to COSMIC v3), and (ii) tested the agreement between our signatures and the DDR gene expression information. For each of the three data sets, we focused on the signatures predicted with an optimum (see *Fitness and Selection*) number of components (11 for BRCA, 5 for PRAD and 3 for OV). For (i), we first transformed our signatures to the standard categories (see Materials and methods) and then computed the cosine similarity between the transformed signature and each of the COSMIC signatures. For (ii), we used CCA to compute the correlation between a sample's signature exposure and its vector of expression of DDR genes. Furthermore, we evaluated the correlation with expression information for subsets of the DDR genes as taken from [22]: Homologous recombination (HR), Base excision repair (BER), Mismatch excision repair (MMR), and Nucleotide excision repair (NER).

In total, 13 of the 19 signatures we discovered were similar to a COSMIC signature (cosine similarity greater than 0.8), spanning 9 distinct COSMIC signatures (1,3,4,5,6,8,12,16,26), the majority of which are known to be active in the studied cancer types. The APOBEC-related COSMIC signatures 2 and 13, that are known to be active in breast cancer, are absent from the signatures extracted using the DDC categorization, which might indicate that this categorization neglects non-DDR mutational processes. Interestingly, a signature similar to COSMIC signature 12 was extracted from both BRCA and OV datasets. The etiology of this signature is unknown and it was previously observed mainly in liver cancer samples. 7 of the 19 signatures were highly correlated (Pearson correlation greater than 0.3) to the expression information. These results are summarized in S1 Table and an example signature (#3) is depicted in Fig 5. This analysis allowed us to compare the suspected activity of each of the 7 signatures with the

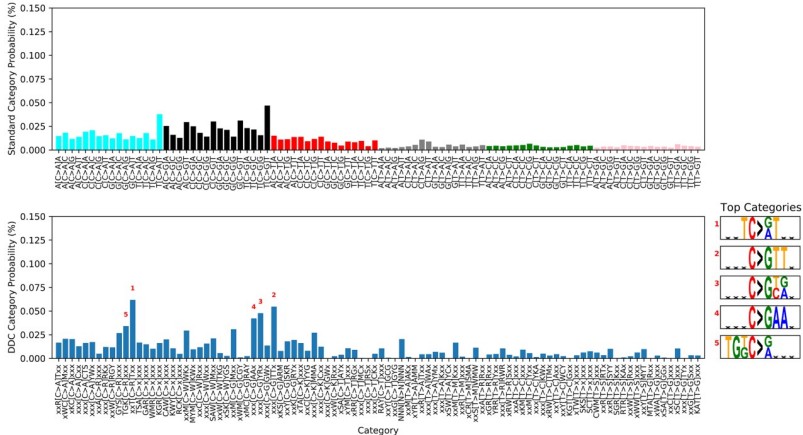

**Fig 5. An example DDC signature (#3 in Table 1).** The signature is depicted using both the standard categories (top) and the DDC ones (bottom, categories presented using the IUPAC code). This signature is similar to COSMIC Signature 3 (cosine similarity 0.89), and its exposure is correlated with the expression of DDR genes (Pearson correlation 0.37).

proposed etiologies of similar COSMIC signatures when such were available. The results are summarized in Table 1. Overall, the gene subsets whose expression was correlated with the exposures of the identified signatures matched the etiologies associated with their COSMIC parallels. For instance, DDC signatures 3 and 6 were correlated with homologous recombination genes and were similar to COSMIC Signature 3, which is associated with HR deficiency. DDC signature 4 was correlated with the expression of base excision repair genes and was similar to COSMIC Signature 1, which is associated with spontaneous deamination of 5-methylcytosine. The correlation to BER genes may be related to their role in the demethylation of 5-mC and the repair of T:G mismatches [24, 25]. DDC signature 5 was most similar to COSMIC Signature 8 and was correlated with the expression of nucleotide excision repair genes. Deficiency of nucleotide excision repair was previously suggested to be associated with COSMIC Signature 8 [26], although the etiology of this signature is still in debate. In two of the cases (signatures 2 and 7, S1 Fig) the COSMIC proposed etiologies did not match the discovered correlations. Finally, DDC signature 1 features a predominance of C-to-T transition, mostly in the G downstream context. While it did not have a clear COSMIC parallel, it was most similar

**Table 1. Signatures that had been discovered using the DDC categorization and were highly correlated (correlation > 0.3) with the expression of DDR genes.**

| DDC # | Similar COSMIC # | Proposed etiologies | Most correlated subset | Most correlated to similar COSMIC |
|-------|------------------|---------------------|------------------------|-----------------------------------|
| 1 | – | – | NER | – |
| 2 | 5,3,8 | HR | BER | BER |
| 3 | 3 | HR | HR | HR |
| 4 | 1 | 5mC deamination | BER | BER |
| 5 | 3,8,5 | HR | NER | MMR |
| 6 | 3,8 | HR | HR | HR |
| 7 | 1,6 | 5mC deamination,MMR | HR | HR |

The signatures are ordered according to these correlations (non-decreasing). For each signature, given are its similar COSMIC signatures (ordered, cosine similarity > 0.8), their COSMIC etiologies if available, and highest correlated DDR gene subset (correlation > 0.2). Additionally, for each DDC signature, we report the most correlated DDR subset to the corresponding similar COSMIC signature.

to the aging-related Signatures 1 and 5 (cosine similarity > 0.75), and exhibited a mild correlation to age (Pearson correlation of 0.2). In addition, for each of the 7 DDC signatures, we tested the correlation between the exposures of the corresponding similar COSMIC signature and the expression of the gene subsets. We observed that the most correlated subsets were in agreement with those identified for the corresponding DDC signatures.

## Conclusion

We have devised a framework for data-driven identification of mutation categories. This framework should serve as a pre-step for mutational signature analysis, and should be applied only periodically when the mutation data base is updated. Applying this framework to breast and prostate cancer whole-exome sequencing data, we identified a novel categorization that leads to better reconstruction of mutation data and higher correlations with the expression patterns of related gene sets over multiple cancer types. A majority of the signatures identified using the data-driven categorization were similar to known COSMIC signatures and were associated with DDR gene subsets that agreed well with the proposed etiologies of their COSMIC parallels. However, several signatures did not have a clear COSMIC parallel, including signatures with high correlation to DDR gene expression. This may indicate that the data-driven categorization, which takes into account broader mutation contexts, uncovers new mutational signatures. Potentially, these signatures could provide insight into new mutational processes, or a more accurate depiction of known processes.

Our categorization process is based on the assumption that the activities of DDR genes are correlated with the mutational processes that are active in a given patient. This assumption does not take into account the changes in gene expression over time since the first mutations occurred. It also downweights non-DDR mutational processes such as processes with environmental causes. Nevertheless, the generalization of the identified categorization to different cancer types may indicate that the approach we take is able to uncover real signals and that mutational processes should be studied with broader mutation contexts than those used to date.

The framework can be easily adapted to find data-driven categorization based upon different assumptions and motivations. For example, the genetic algorithm fitness function could be modified to account for correlation with other gene sets and factorization reconstruction error, or to prefer high mutational signature entropy to improve interpretability.

Our work focused on 96 categories—the number of standard categories—in order to eliminate the number of categories in a categorization as a possible explanation for differences to previous work. However, as this number is arbitrary, it would be interesting to investigate (i) the possibility of decreasing the number of categories while maintaining comparable performance, and (ii) whether significant improvements in performance could be obtained by increasing the number of categories. To test these ideas, we suggest running our method with a flexible number of categories per categorization. For example, a simple adjustment to our method would be to introduce category addition and deletion mutations in each generation.

## Supporting information

**S1 Fig. An example of a non-flat DDC signature (#7 in Table 1).** The signature is depicted using both the standard categories (top) and the DDC ones (bottom, categories presented using the IUPAC code). This signature is similar to COSMIC Signatures 1 and 6 (cosine similarity 0.88 and 0.84 respectively), and its exposure is correlated with the expression of DDR genes (Pearson correlation 0.30).
(PDF)

**S2 Fig. Comparative performance evaluation: Correlation to expression of odorant binding genes (used as a control set that is not expected to be related to mutational signature activities).** For each $K$ in the range of number of components [2, max($k\_cosmic$, $K^*$) + 2], we apply NMF to the WGS samples to learn the signature matrix $H$ and then derive the exposure matrix $W$ from the WES samples using NNLS. We learn the CCA coefficients using WES training samples and compute the resulting correlation on the test samples. The reported correlation is the average over 10-fold cross validation. Error bars represent the standard deviation of multiple evaluation runs.
(PDF)

**S3 Fig. Comparative performance evaluation by reconstruction error (log-transformed).** For each $K$ in the range of number of components [2, max($k\_cosmic$, $K^*$) + 2], we apply NMF to the WGS samples to learn the signature matrix $H$ and then derive the exposure matrix $W$ from the test samples using NNLS. The reconstruction error (Kullback–Leibler divergence) is the approximation error of this factorization with respect to the test samples of the (normalized) count matrix $V$. NMF is applied 10 times and the reported reconstruction error is the average over these runs. Error bars represent the standard deviation.
(PDF)

**S1 Table. All 19 signatures that had been discovered using the DDC categorization on the BRCA, PRAD and OV datasets.** The signatures are ordered according to their correlation with the expression of DDR genes. For each signature, given are also the dataset that it was derived from, its similar COSMIC signatures (ordered, cosine similarity > 0.8) and the similarity to the most similar COSMIC signature. In case there is no COSMIC signature with cosine similarity > 0.8 to the DDC signature, we report the COSMIC signature with the greatest similarity to it.
(JPG)

## Acknowledgments

We thank Itay Sason for his critical reading of the manuscript.

## Author Contributions

**Conceptualization:** Gal Gilad, Mark D. M. Leiserson, Roded Sharan.

**Data curation:** Gal Gilad.

**Funding acquisition:** Mark D. M. Leiserson, Roded Sharan.

**Methodology:** Gal Gilad, Mark D. M. Leiserson, Roded Sharan.

**Resources:** Roded Sharan.

**Software:** Gal Gilad.

**Supervision:** Roded Sharan.

**Writing – original draft:** Gal Gilad, Roded Sharan.

**Writing – review & editing:** Mark D. M. Leiserson.

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
