## [Decision Letter · Decision Letter 0]

26 Jul 2021

Dear Gilad,

Thank you very much for submitting your manuscript "A data-driven approach for constructing mutation categories for mutational signature analysis" for consideration at PLOS Computational Biology.

As with all papers reviewed by the journal, your manuscript was reviewed by members of the editorial board and by several independent reviewers. In light of the reviews (below this email), we would like to invite the resubmission of a significantly-revised version that takes into account the reviewers' comments.

We cannot make any decision about publication until we have seen the revised manuscript and your response to the reviewers' comments. Your revised manuscript is also likely to be sent to reviewers for further evaluation.

Sincerely,

Pedro Mendes, PhD

Associate Editor

PLOS Computational Biology

Sushmita Roy

Deputy Editor

PLOS Computational Biology

Reviewer's Responses to Questions

**Comments to the Authors:**

Reviewer #1: The authors present newly developed methodology for mutational signature extraction that takes into account flexibly expanded nucleotide context and matched DDR gene activity. The rationale of searching for patterns that reflect similar DDR processes is reasonable and one would indeed expect the resulting signatures to have better interpretability. The generation of these signatures employs a genetic algorithm which has been adapted for the question but whose components are not entirely motivated. Overall, while the idea of combining mutation and DDR expression information for signature discovery is very interesting, the manuscript lacks transparency in the uncovered signatures, their overall interpretability and applicability.

I have the following major concerns:

• The method employs a genetic algorithm that is based on crossover and mutation. It is unclear to me what relevance/interpretation the crossover would have in this context, given that we are not talking about sexual reproduction in the context of tumour cell proliferation. I understand the “parents” in this case are abstracted to refer to different mutation categories, but why would there be a crossover in the process of cell division?

• Why 96 categories? This is not explained, and I cannot think of why it may be beneficial to replicate the 96-category standard given that the categories are being changed.

• It is somewhat unclear what is going on when exposures and expression are being correlated. Are there any correlation cut-offs applied to guide signature selection? These correlations should also be plotted/reported, as they would enhance the interpretability of the procedure.

• For the calculation of the reconstruction error, 10 runs with randomly initialised W,H are performed and the smallest error obtained is reported. It may be fairer to report the mean and confidence interval instead – and this could be updated in Fig 3.

• Figure 4: DDC improvement is very marginal in BRCA - any particular interpretation as to why this is the case?

• Table 1: it would be useful to discuss further the fact that more often than not the most correlated subset and proposed aetiology do not match. I acknowledge part of this is covered in the discussion already (BER and deamination connection), but it may also be worth considering that 4/7 top signatures are essentially flat signatures. Would there be a tendency for flat signatures to naturally display the highest correlations? Is Pearson’s correlation the best measure of correlation, particularly when some contributions will be so small?

• In addition to this, it may be useful to explore further, or at least display, some of the non-flat signatures - possibly a signature correlated with an MMR-deficiency signature like SBS6?

• All signatures derived in the analysed cohorts should be reported, along with the corresponding DDR correlations.

• Do the signatures uncovered make sense in the context of the biology of the three diseases studied, and that of other reports from the literature? Are there any new biological insights that this analysis brings?

• The authors should expand on the motivation, interpretability and applicability of their newly uncovered signatures. Can these signatures improve DDR deficiency prediction? Other benefits and future directions should be specified.

Reviewer #2: The authors present a new method that uses a genetic algorithm to identify fine-scale latent signatures of mutation processes in cancer genomes. Much of the recent development in mutation signature analysis has stagnated on minor tweaks to the non-negative matrix factorization and focused primarily on signatures defined by trinucleotide mutation context, so it is exciting to see new algorithm developments in this area and consideration for broader sequence contexts. I also thought the use of gene expression data from DNA damage+repair genes was a clever idea for validating the mutation signatures detected with the DDC method. However, I am concerned that the implementation of this method is prohibitively computationally costly, and the authors did not provide sufficient evidence to demonstrate that this method will lead to new insights into mutation processes and cancer biology.

MAJOR COMMENTS

The authors state that their algorithm took 2 days to run on a 40-core server, and the description of the tuning procedure indicates that they ran a total of 4+3+3+4=14 iterations to explore different values of the model parameters, which suggests the tuning process took 4 weeks of compute time. I appreciate that this is a new approach to mutation signature analysis and optimization is not necessarily a priority, but I’m not convinced that the novelty of the findings justifies this massive computational burden. Given the length of the paper and focus on describing the algorithm, I reviewed this primarily as a method/software paper, and as such, I would expect the software to have a runtime at least on the same order of magnitude as existing methods (The authors state that they identified a categorization that generalizes across datasets [line 320], so at the very least I would expect these data to be made publicly available so other users can treat it as a “pre-trained” model). On the other hand, if this is meant to be more of a standalone research paper, the biological insights lack novelty/depth. For example, Table 1 indicates DDC signature 1 is the only signature without a similar COSMIC signature or proposed etiology, but is most correlated to NER genes and mildly correlated with age. How do we know this is not just an artifact of fitting to the data or a spurious correlation? Is there any evidence in prior literature that NER genes or age-related processes are enriched for C>T transitions in the G downstream context (line 407)?

It would also be nice to include:

- A supplementary figure comparing performance across the different parameter values evaluated (similar to Figs 3 & 4 but comparing e.g., q = [0, 0.4, 0.8, 1])

- scripts in the Github repository so users can compare performance of DDC on their data against standard approaches

Line 303-307: The authors state that they selected COSMIC signatures only found to be active in the investigated cancer types. It would be good to include some supplementary results comparing the DDC signatures against all COSMIC signatures to affirm that the matching signatures in Table 1 are not just reflecting a confirmation bias. Moreover, the COSMIC signatures believed to be associated with sequencing artifacts should absolutely be considered in this analysis (https://cancer.sanger.ac.uk/signatures/sbs/).

The authors state that their choice to collapse longer sequence contexts into the 96 most informative categories was solely for the sake of comparing their findings directly with the standard 96-subtype approaches, not by any biological motivation. It would be nice to see a more thorough analysis of how many categories are actually needed--what is the minimal number of categories required to match the reconstruction error of standard approaches? How much is reconstruction error reduced by increasing the number of categories? Do the identified categories remain fairly stable?

I had some difficulty getting the DDC software to run. It would help to include a conda/pip environment config file to ensure users are always using tested versions of the required Python libraries (using a fresh install of Miniconda, I was able to run the software after installing the scikit-learn, matplotlib, pandas, and pyrcca packages). I also got several numpy runtime warnings while running DDC, but it was not obvious if this was informational or an error that needed to be debugged (e.g., “python3.9/site-packages/numpy/lib/function_base.py:380: RuntimeWarning: Mean of empty slice.”). To run the software with the included example data, I had to create a new directory called “categorization_related_data”, move the example files to this directory, and rename them to remove underscores in the file names, because the “--datatype” parameter treats underscores as a delimiter. The code is cleanly written and easy to follow, but additional testing and documentation would go a long way towards making this software functional.

MINOR COMMENTS

Line 35-39--this sentence should include some references to the initial implementations of NMF-based mutation signature analysis using 96 categories (Nik-Zainal et al. 2012 [https://www.sciencedirect.com/science/article/pii/S0092867412005284] and Alexandrov et al. 2013 [https://www.nature.com/articles/nature12477])

Line 75, Line 168, Fig. 1--the terminology of “8-long sequence” is very confusing. Functionally, the authors are describing how they encode a 7-mer mutation subtype as an 8-character string (e.g., TGT[C>T]AAC becomes TGTCTAAC), but calling the latter a “sequence” implies they are looking at an actual 8bp DNA sequence motif, which is misleading. It would be clearer to describe the mutation categories as representing a possible 7-mer mutation subtype and us standard IUPAC codes (http://www.bioinformatics.org/sms/iupac.html) to represent “wildcards” present in the categories, e.g., NNTC(A/G)GAN would be better represented as NNT[C>R]GAN, where R represents an A or a G. This way, all categories are consistently written in a way that indicates the unmutated flanking bases and the specific mutation at the 4th position. (I’ll also note that the authors already use the “7-mer” terminology on the README of their Github repository, and the python code is already using IUPAC notation for ambiguous bases, but the --kmer parameter in the program is inconsistent with this).

Fig. 5--the DDC signature panel would be easier to interpret if the signature barplots indicated which bars correspond to each of the “Top Categories” in the logo plots, e.g., with a number 1-5 above the corresponding bar or color code for each category.

Reviewer #3: The paper presents a data-driven approach to extract mutation categories. Such mutational categories can be used to analyze mutational signatures in cancer going beyond the standard and arbitrary choice of the "canonical" mutational signatures.

While the paper does propose a novel methodology to extract mutation categories, the paper's main contributions are the analysis and the results. The methodology is sound, and the results are interesting. The fact that the resulting categorizations are variable in size and greatly deviate from the standard categories is extremely interesting, and may inspire other approaches to develop data-driven signatures.

The main assumption, that is, tumor cells that are expose to similar mutational processes show similar expression levels of genes that are involved in the underlying mutagenic process may be fairly strong and not always satisfied. However, I think it is a reasonable assumption, and maybe even required, to develop a data-driven approach.

I do not have any major request for edit. As a minor, at the beginning of Section 2 it would be good to mention, as in line 185, that the focus on 96 categories is to compare with the standard categorization. Moreover, do the author have any thoughts on what would be the best choice of the number of categories? This could be added to Conclusions.

**Have the authors made all data and (if applicable) computational code underlying the findings in their manuscript fully available?**

Reviewer #1: Yes

Reviewer #2: **No: **Missing code for comparing against COSMIC signatures; missing data for the DDC categories & resulting signatures

Reviewer #3: Yes

PLOS authors have the option to publish the peer review history of their article (what does this mean?). If published, this will include your full peer review and any attached files.

Reviewer #1: No

Reviewer #2: No

Reviewer #3: No
---

## [Decision Letter · Decision Letter 1]

6 Oct 2021

Dear Gilad,

We are pleased to inform you that your manuscript 'A data-driven approach for constructing mutation categories for mutational signature analysis' has been provisionally accepted for publication in PLOS Computational Biology.

Best regards,

Pedro Mendes, PhD

Associate Editor

PLOS Computational Biology

Sushmita Roy

Deputy Editor

PLOS Computational Biology

Reviewer's Responses to Questions

**Comments to the Authors:**

Reviewer #1: The authors have addressed my concerns in a satisfactory manner. I have no further comments.

Reviewer #2: The authors have sufficiently addressed all of my previous comments.

**Have the authors made all data and (if applicable) computational code underlying the findings in their manuscript fully available?**

Reviewer #1: Yes

Reviewer #2: Yes

PLOS authors have the option to publish the peer review history of their article (what does this mean?). If published, this will include your full peer review and any attached files.

Reviewer #1: No

Reviewer #2: No

---

## [Editor Report · Acceptance letter]

15 Oct 2021

PCOMPBIOL-D-21-00767R1 

A data-driven approach for constructing mutation categories for mutational signature analysis

Dear Dr Gilad,

I am pleased to inform you that your manuscript has been formally accepted for publication in PLOS Computational Biology. Your manuscript is now with our production department and you will be notified of the publication date in due course.

With kind regards,

Anita Estes
